# CPT: COLORFUL PROMPT TUNING FOR PRE-TRAINED VISION-LANGUAGE MODELS

## ABSTRACT

Pre-Trained Vision-Language Models (VL-PTMs) have shown promising capabilities in grounding natural language in image data, facilitating a broad variety of cross-modal tasks. However, we note that there exists a significant gap between the objective forms of model pre-training and fine-tuning, resulting in a need for large amounts of labeled data to stimulate the visual grounding capability of VL-PTMs for downstream tasks. To address the challenge, we present Cross-modal Prompt Tuning (CPT, alternatively, Colorful Prompt Tuning), a novel paradigm for tuning VL-PTMs, which reformulates visual grounding into a fill-in-the-blank problem with color-based co-referential markers in image and text, maximally mitigating the gap. In this way, CPT enables strong few-shot and even zero-shot visual grounding capabilities of VL-PTMs. Comprehensive experimental results show that the prompt-tuned VL-PTMs outperform their fine-tuned counterparts by a large margin (e.g., 17.3% absolute accuracy improvement, and 73.8% relative standard deviation reduction on average with one shot in RefCOCO evaluation). All the data and codes will be available to facilitate future research.

## 1 INTRODUCTION

Grounding natural language in fine-grained image regions is essential for a broad variety of vision-language tasks, such as robotic navigation (Tellex et al., 2011; Anderson et al., 2018b), visual question answering (Antol et al., 2015; Anderson et al., 2018a), visual dialogue (Das et al., 2017), and visual commonsense reasoning (Zellers et al., 2019). Recently Pre-Trained Vision-Language Models (VL-PTMs) have shown promising capabilities in visual grounding. Typically, generic cross-modal representations are first pre-trained on large-scale image-caption data in a self-supervised fashion, and then fine-tuned to adapt to downstream tasks (Lu et al., 2019; Su et al., 2019; Li et al., 2020; Radford et al., 2021). This *pre-training-then-fine-tuning* paradigm of VL-PTMs has greatly pushed forward the state-of-the-art of many cross-modal tasks.

Despite the success, we note that there exists a significant gap between the objective forms of pre-training and fine-tuning of VL-PTMs. As illustrated in Figure 1, during pre-training, most VL-PTMs are optimized based on the masked language modeling objective, trying to recover the masked token from the cross-modal context. However, during fine-tuning, downstream tasks are usually conducted by classifying unmasked token representations into semantic labels, where task-specific parameters are typically introduced. The gap hinders the effective adaptation of VL-PTMs to downstream tasks. As a result, a large amount of labeled data is typically required to stimulate the visual grounding capabilities of VL-PTMs for downstream tasks.

In this work, inspired by recent progress in pre-trained language models in natural language processing (Brown et al., 2020; Schick & Schütze, 2021a; Liu et al., 2021), we present Cross-modal Prompt Tuning (CPT, alternatively, Colorful Prompt Tuning), a novel paradigm for tuning VL-PTMs. The key insight is that by adding color-based co-referential markers in both image and text, visual grounding can be reformulated into a fill-in-the-blank problem, maximally mitigating the gap between pre-training and fine-tuning. As shown in Figure 1, to ground natural language expressions in image data, CPT consists of two components: (1) a *visual sub-prompt* that uniquely marks image regions with colored blocks or segmentation masks, and (2) a *textual sub-prompt* that puts the query text into a color-based query template. Explicit grounding to the target image region can then be achieved by recovering the corresponding color text from the masked token in the query

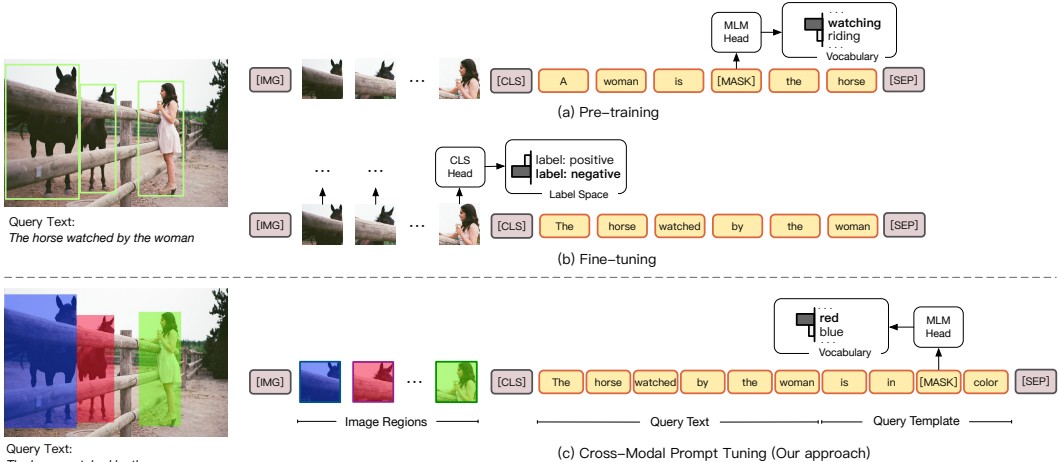

Figure 1: Illustration of (a) pre-training for VL-PTMs with masked language modeling (MLM) head, (b) vanilla fine-tuning with new classification (CLS) head, and (c) our colorful cross-modal prompt tuning (CPT) framework that reformulates visual grounding into a fill-in-the-blank problem with reused MLM head. Only square parts of relevant image regions are shown for illustration.

template. In addition, we present a principled method to search for high-quality cross-modal prompt configurations (i.e., visual appearances and texts of colors) for CPT.

By mitigating the gap from pre-training, CPT enables strong few-shot and even zero-shot visual grounding capabilities of VL-PTMs. Experimental results show that the prompt-tuned VL-PTMs outperform their fine-tuned counterparts by a large margin. For example, using colored blocks as visual sub-prompts, CPT achieves $17.3\%$ absolute accuracy improvement, and $73.8\%$ relative standard deviation reduction on average with one shot in RefCOCO evaluation. In the same setting, when equipped with colored segmentation masks as visual sub-prompts, CPT can further achieve $20.0\%$ absolute accuracy improvement, and $76.2\%$ relative standard deviation reduction than the vanilla fine-tuning approach.

Our contributions are summarized as threefold: (1) We present a novel cross-modal prompt tuning paradigm for VL-PTMs. To the best of our knowledge, this is the first attempt in both cross-modal prompt tuning for VL-PTMs, and zero- and few-shot visual grounding independent of object types. (2) We present a principled approach to search for high-quality cross-modal prompt configurations for CPT. (3) We conduct comprehensive experiments which demonstrate the effectiveness of CPT.

## 2    PRELIMINARY

In the literature, visual grounding is typically formulated as a referring expression comprehension (REC) problem (Plummer et al., 2015; Mao et al., 2016). Given an image $I$ and a query text of referring expression $q$, REC aims to locate the target region in $I$ that corresponds to $q$. In this section, we introduce the vanilla fine-tuning approach for VL-PTMs.

A common practice for REC is to first detect a set of region proposals $\{v_1, v_2, \ldots, v_n\}$ via object detectors, and then classify or rank the proposals to select the target region (Lu et al., 2019; Chen et al., 2020). Specifically, visual and textual inputs are first transformed into a sequence of input tokens $\{[\text{IMG}], v_1, v_2, \ldots, v_n, [\text{CLS}], w_1, w_2, \ldots, w_m, [\text{SEP}]\}$, where $\{w_1, w_2, \ldots, w_m\}$ are textual tokens of $q$, and $[\text{IMG}]$, $[\text{CLS}]$ and $[\text{SEP}]$ are special tokens. To obtain input representations, the feature of image regions is extracted by visual encoders, and the embeddings of textual and special tokens are obtained by a lookup table. Then input representations are fed into the pre-trained transformers to produce the hidden representations $\{\mathbf{h}_{[\text{IMG}]}, \mathbf{h}_v^1, \mathbf{h}_v^2, \ldots, \mathbf{h}_v^n, \mathbf{h}_{[\text{CLS}]}, \mathbf{h}_w^1, \mathbf{h}_w^2, \ldots, \mathbf{h}_w^m, \mathbf{h}_{[\text{SEP}]}\}$. Finally the hidden representation of the target region is optimized against negative ones via classification or ranking loss, where new task-specific parameters are introduced. As a result, fine-tuned VL-PTMs need a large mount of labeled instances to stimulate the visual grounding capability.

# 3   CROSS-MODAL PROMPT TUNING (CPT)

In this section, we introduce the framework of CPT, and how to apply CPT to zero-shot, few-shot and fully supervised visual grounding.

## 3.1   OVERVIEW

The key to visual grounding is to establish fine-grained connections between image regions and textual expressions. Therefore, a good cross-modal prompt tuning framework should take full advantage of co-referential signals from both image and text, and maximally mitigate the gap between pre-training and tuning. To this end, CPT reformulates visual grounding into a fill-in-the-blank problem, as shown in Figure 1. Specifically, the CPT framework consists of two components: (1) a *visual sub-prompt* that uniquely marks the image regions with colored blocks or segmentation masks, and (2) a *textual sub-prompt* that puts the query text into a color-based query template. Equipped with CPT, it is then straightforward for VL-PTMs to ground the query text by filling the masked token with the color text of the target image region, where the objective form is identical to pre-training.

## 3.2   VISUAL SUB-PROMPT

Given an image $I$ and its region proposals $\mathcal{R} = \{v_1, v_2, \ldots, v_n\}$, visual sub-prompt aims to uniquely mark the image regions with natural visual makers. Interestingly, we note that colored bounding boxes are widely used to uniquely mark objects in images *for visualization* in the literature. Inspired by this, we bridge the image regions and query text through a set of colors $\mathcal{C}$, where each color $c_i = (c_v^i, c_w^i) \in \mathcal{C}$ is defined by its visual appearance $c_v^i$ (e.g., RGB (255, 0, 0)) and color text $c_w^i$ (e.g., *red*). Then we mark each region proposal $v_i$ in the image with a unique color $c_v^i$ for grounding, resulting in a set of colored image proposals $\Psi(\mathcal{R}; \mathcal{C})$, where $\Psi(\cdot)$ denotes visual sub-prompt.

As for the shape of the visual sub-prompt, in principle, there are multiple plausible choices to mark the regions with colors, including colored bounding boxes, solid blocks, or solid object segmentation masks. In our experiments, we find that coloring the object with solid blocks and segmentation masks yields better results than bounding boxes, since solid colors that fit the outlines of objects are more common in real-world images (e.g., *red shirt* and *blue car*). Note that the addition of visual sub-prompt to the raw image does not change the architecture or parameters of VL-PTMs.

## 3.3   TEXTUAL SUB-PROMPT

Textual sub-prompt aims to prompt VL-PTMs to establish the connections between the query text and image regions marked by visual sub-prompt. Specifically, the query text $q$ (e.g., "*the horse watched by the woman*") is transformed into a fill-in-the-blank query using a template $\mathcal{T}_g(\cdot)$ as:

$$\mathcal{T}_g(q) = \texttt{[CLS]} \; q \text{ is in } \texttt{[MASK]} \text{ color } \texttt{[SEP]}$$

In this way, VL-PTMs are prompted to decide the color of which region is more appropriate to fill in the mask (e.g., *red* or *blue*) as follows:

$$P(v = v_i | \mathcal{R}, q) = P(\texttt{[MASK]} = c_w^i | \Psi(\mathcal{R}; \mathcal{C}), \mathcal{T}_g(q)) = \frac{\exp(\mathbf{h}_{\texttt{[MASK]}}^\top \mathbf{c}_w^i)}{\sum_{c_j \in \mathcal{C}} \exp(\mathbf{h}_{\texttt{[MASK]}}^\top \mathbf{c}_w^j)}, \qquad (1)$$

where $v$ is the target region, $\mathbf{c}_w^i$ is the embedding of $c_w^i$ in the pre-trained MLM head. Note that the procedure does not introduce any new parameters, and also mitigates the gap between pre-training and tuning, and therefore improves the data efficiency for tuning VL-PTMs.

## 3.4   TRAINING AND INFERENCE

Equipped with CPT, VL-PTMs can readily perform zero-shot visual grounding without any labeled data, since the cross-modal representations of colors and their composition with other concepts (e.g., objects, attributes and relations) have been well learned by VL-PTMs during pre-training. When a few or full labeled instances are available, VL-PTMs can be further tuned by CPT using the entropy-based objective: $\mathcal{L} = -\sum_{(\mathcal{R}, q, v^\star) \in \mathcal{D}_{\text{train}}} \log P(v^\star | \mathcal{R}, q)$, where $\mathcal{D}_{\text{train}}$ is the training set.

Figure 2: CPT framework for predicate classification by filling-in-the-blank with reused MLM head.

Although it is appealing to bridge the image and text through a color-based prompt, we identify two key challenges in its design: (1) how to determine the configurations of the color set $\mathcal{C}$, and (2) how to deal with the large number of image regions with limited pre-trained colors.

**Cross-Modal Prompt Search.** Previous works in textual prompt tuning show that prompt configurations (e.g., textual templates) have a significant influence on the performance (Jiang et al., 2020). In this work, we make the first investigation in searching the cross-modal prompt configuration (i.e., the color set $\mathcal{C}$). Intuitively, $\mathcal{C}$ should consist of colors to which VL-PTMs are the most sensitive. To obtain a color $c_i = (c_v^i, c_w^i)$, a naive approach is to adopt the most frequent color text in the pre-training text as $c_w^i$, and its standard RGB as $c_v^i$ (e.g., $c_i = ((255, 0, 0), red)$). However, this solution is sub-optimal, since it determines the color text without considering its visual appearance, and the visual appearance of a color in real-world images often differs from its standard RGB.

To address the challenge, we present a principled cross-modal prompt search (CPS) algorithm for CPT, which jointly considers visual and textual semantics in real-world cross-modal data. Specifically, we first identify a candidate set of color texts $\hat{\mathcal{C}}_w$ and visual appearances $\hat{\mathcal{C}}_v$. For each visual appearance candidate $\hat{c}_v \in \hat{\mathcal{C}}_v$, we feed into VL-PTMs a pseudo-data instance consisting of a pure colored block of $\hat{c}_v$ and a text: "[CLS] a photo in [MASK] color [SEP]". Then we compute the decoding score $s(\hat{c}_v, \hat{c}_w)$ for each color text candidate $\hat{c}_w \in \hat{\mathcal{C}}_w$ as in Equation 1, where a larger decoding score indicates higher correlation between $\hat{c}_v$ and $\hat{c}_w$. To select the color texts that are sensitive by VL-PTMs, we retain the color texts that achieve the largest decoding scores for visual appearance candidates: $\mathcal{C}_w = \{c_w | c_w = \arg\max_{\hat{c}_w^j \in \hat{\mathcal{C}}_w} s(\hat{c}_v^i, \hat{c}_w^j), \hat{c}_v^i \in \hat{\mathcal{C}}_v\}$. Similarly, we can obtain the visual appearances according to the largest decoding score, resulting in the color set: $\mathcal{C} = \{(c_v, c_w) | c_v = \arg\max_{\hat{c}_v^i \in \hat{\mathcal{C}}_v} s(\hat{c}_v^i, c_w^j), c_w^j \in \mathcal{C}_w\}$. We refer readers to Section B for the pseudo-code of the algorithm. In experiments, we find that the resultant colors yield better results than the naive ones. To make the raw content of the colored image regions available to VL-PTMs, a transparency hyperparameter $\alpha \in (0, 1)$ is further applied to color visual appearances in practice.

**Image Region Batching.** In visual grounding, the number of region proposals in an image usually exceeds the size of $\mathcal{C}$ ($\sim 10$). Besides, we observe that heavily overlapped colored blocks can hinder visual grounding. Therefore, we divide the image regions into batches, where each batch contains a handful of moderately overlapping image regions, and mark each batch with a visual sub-prompt respectively. To handle the batches that do not contain the target region, we further introduce a new candidate text *none* in the decoding vocabulary, to indicate that there is no target region in the batch.

## 3.5 CPT FOR PREDICATE CLASSIFICATION

In the previous sections, we introduced CPT for visual grounding. In fact, CPT can also be easily adapted to other cross-modal tasks, such as predicate classification. Given an object pair (including the categories and bounding boxes) in an image, predicate classification aims to classify the relation into a relation set $\mathcal{P}$, providing structured image representations that can facilitate many cross-modal tasks (Johnson et al., 2015; Hudson & Manning, 2019; Shi et al., 2019). In the literature, since the ground-truth relations cannot be exhaustively annotated during evaluation, to avoid false negatives, previous works typically score the triplets and evaluate the recall of top-N triplets (Xu et al., 2017; Zellers et al., 2018; Chen et al., 2019; Tang et al., 2019).

**Visual and Textual Sub-prompts.** As shown in Figure 2, to perform predicate classification, CPT first marks the image regions with visual sub-prompt as in Section 3.2, and puts the object pair in the query template as follows:

$$\mathcal{T}_r(s, o) = \text{[CLS] The } s_w \text{ in } c_w^i \text{ color is [MASK] the } o_w \text{ in } c_w^j \text{ color [SEP]}$$

where $s_w$ is the subject text, $o_w$ is the object text, and $c_w^i$ and $c_w^j$ are the corresponding color texts. Then VL-PTMs are prompted to recover the relation texts from masked tokens in the template. To

accommodate the varied number of tokens in relation texts (e.g., *wearing*, *walking on*, typically 1∼3 tokens), we introduce a variable $l$ indicating the number of tokens in a relation text (e.g., $l = 2$ for *walking on*). The template $\mathcal{T}(\cdot; l)$ will have $l$ consecutive masked tokens for relation prediction. For each template $\mathcal{T}(\cdot; l)$, we introduce a special NA relation consisting of $l$ tokens, which indicates that there is no relation between the entity pair under $\mathcal{T}(\cdot; l)$. Specifically, in our experiments, the NA relation is *irrelevant*, *no relation*, *no relation with* for $l = 1, 2, 3$ respectively.

**Training**. Given a relational triplet $(s, r, o)$, after decorating the input image regions and the object pair with visual and textual sub-prompts, VL-PTMs are optimized with the MLM loss to recover the relational tokens. Specifically, denote the number of tokens in $r$ as $|r|$. (1) For templates where $l = |r|$, models are asked to reconstruct the $i$th masked token in $\mathcal{T}(s, o; l)$ with the $i$th relational token $r_i$ using the MLM head. (2) For templates where $l \neq |r|$, since there is no relation between $(s, o)$ under $\mathcal{T}(s, o; l)$, models are asked to reconstruct the NA relation. For $(s, o)$ that do not have any relation in the image, models are asked to reconstruct the NA relation for all $\mathcal{T}(s, o; l)$.

**Inference**. During inference, given an object pair $(s, o)$, we score the relations based on their fitness to the prompt context. Specifically, the score of each relation $r \in \mathcal{P} \cup \{\text{NA}\}$ is obtained by the aggregated MLM scores of its composing tokens under the corresponding template: $s(r) = \frac{1}{l} \sum_{i=1}^{l} \log P(\,[\text{MASK}]_i = r_i | \mathcal{T}(s, o; l))$, where $l = |r|$. Intuitively, larger $s(r)$ indicates that the relation $r$ better fits the prompt context. Finally, the triplets $(s, r, o)$ are ranked according to the relation score $s(r)$, where $r \in \mathcal{P}$.

Compared with visual grounding that aims to locate image regions for ungrounded texts, predicate classification represents a different series of cross-modal tasks that aim to perform semantic recognition based on grounded inputs, such as object classification (Zhao et al., 2017) and scene graph classification (Xu et al., 2017). In addition to better data efficiency, a crucial advantage of using CPT is that the semantic labels can be produced from open-world vocabularies, instead of fixed label sets.

# 4 EXPERIMENTS

In this section, we empirically evaluate CPT in prompting VL-PTMs for visual grounding in different settings, including zero-shot, few-shot and fully supervised settings. We refer readers to Section C for the implementation details.

## 4.1 EXPERIMENTAL SETTINGS

We first introduce the experimental settings of the visual grounding task, including datasets, training settings, evaluation protocols and baseline models in our experiments.

**Datasets.** Following previous works (Rohrbach et al., 2016; Zhang et al., 2018), we adopt three widely used visual grounding datasets collected from MSCOCO images (Lin et al., 2014), including RefCOCO (Yu et al., 2016), RefCOCO+ (Yu et al., 2016) and RefCOCOg (Mao et al., 2016). We refer readers to Section D.2 for more dataset details.

**Training Settings.** We report experimental results of different training settings, including (1) zero-shot setting, where no training data is available, (2) few-shot setting, where $K$ training instances are available ($K = 1, 2, 4, 8, 16$), and (3) fully supervised setting, where the full training set is available.

**Evaluation Protocols.** (1) Evaluation metrics. Following Zhang et al. (2018); Lu et al. (2019), we adopt accuracy of the grounding results as the evaluation metrics. An expression is considered correctly grounded if the IoU of the top predicted region and the ground truth is greater than $0.5$. (2) Model validation. To better approximate the few-shot scenario where only a few labeled instances are available, inspired by Gao et al. (2021), we use a few-shot validation set (consisting of 16 instances) for few-shot and zero-shot experiments, and use full validation set for fully supervised experiments. (3) Robust evaluation. Previous works have shown that model training on limited data can suffer from instability (Dodge et al., 2020; Gao et al., 2021). For a robust and comprehensive evaluation, we report mean results over 5 random training set splits, as well as the standard deviation. For fair comparisons, the training and validation sets are identical for our baselines and CPT.

**Baselines.** We evaluate two variants of CPT, including CPT using colored blocks (CPT-Blk) and colored segmentation masks (CPT-Seg). We adopt the widely used VinVL (Zhang et al., 2021)

Table 1: Main results. Accuracies (%) of grounding referring expressions in zero-shot, few-shot and fully supervised settings. We report mean and standard deviation performance over 5 random splits. ZS: zero-shot. Blk: colored block, Seg: colored segmentation mask.

| | Shot | Model | RefCOCO | | | RefCOCO+ | | | RefCOCOg | |
|---|---|---|---|---|---|---|---|---|---|---|
| | | | val | testA | testB | val | testA | testB | val | test |
| ZS | 0 | Random | 15.9 ± 0.2 | 19.4 ± 0.6 | 13.4 ± 0.4 | 16.1 ± 0.1 | 13.3 ± 0.6 | 20.0 ± 0.2 | 18.8 ± 0.4 | 19.2 ± 0.3 |
| | | CPT-Blk | 26.9 | 27.5 | 27.4 | 25.4 | 25.0 | 27.0 | 32.1 | 32.3 |
| | | CPT-Seg | **32.2** | **36.1** | **30.3** | **31.9** | **35.2** | **28.8** | **36.7** | **36.5** |
| Few-Shot | 1 | Fine-tuning | 16.5 ± 4.9 | 12.0 ± 6.6 | 23.5 ± 5.7 | 22.2 ± 7.6 | 20.6 ± 9.3 | 25.7 ± 5.2 | 26.9 ± 8.4 | 26.9 ± 8.1 |
| | | CPT-Blk | 34.1 ± 1.3 | 37.7 ± 1.7 | 32.2 ± 1.5 | 35.9 ± 4.1 | 40.4 ± 5.4 | 32.2 ± 2.6 | 39.7 ± 3.4 | 39.9 ± 3.0 |
| | | CPT-Seg | **37.2 ± 0.9** | **41.5 ± 1.5** | **33.2 ± 1.7** | **37.9 ± 4.0** | **42.3 ± 5.9** | **33.9 ± 2.4** | **43.1 ± 2.9** | **43.4 ± 3.1** |
| | 2 | Fine-tuning | 22.5 ± 4.5 | 21.0 ± 7.2 | 25.9 ± 4.7 | 27.0 ± 3.1 | 27.8 ± 4.2 | 27.0 ± 2.6 | 28.4 ± 12.0 | 28.1 ± 11.3 |
| | | CPT-Blk | 35.3 ± 3.2 | 39.6 ± 3.0 | 30.9 ± 1.7 | 33.3 ± 3.6 | 37.5 ± 4.8 | 30.3 ± 2.5 | 40.1 ± 5.1 | 40.0 ± 4.7 |
| | | CPT-Seg | **39.8 ± 1.7** | **45.6 ± 3.2** | **33.9 ± 0.4** | **38.6 ± 3.6** | **44.5 ± 4.5** | **32.8 ± 3.8** | **44.7 ± 5.1** | **44.3 ± 4.8** |
| | 4 | Fine-tuning | 29.1 ± 5.0 | 29.9 ± 7.8 | 29.8 ± 5.3 | 34.2 ± 4.2 | 37.7 ± 5.2 | 30.5 ± 3.3 | 34.0 ± 13.1 | 33.7 ± 12.8 |
| | | CPT-Blk | 38.3 ± 2.1 | 43.6 ± 3.3 | 34.0 ± 1.6 | 38.8 ± 3.8 | 44.4 ± 6.4 | 33.5 ± 1.5 | 40.6 ± 7.9 | 40.9 ± 7.9 |
| | | CPT-Seg | **40.7 ± 3.2** | **47.4 ± 4.1** | **35.3 ± 1.8** | **40.3 ± 2.0** | **46.5 ± 3.1** | **34.5 ± 1.5** | **44.4 ± 6.9** | **44.4 ± 6.9** |
| | 8 | Fine-tuning | 34.6 ± 4.8 | 37.8 ± 5.5 | 31.4 ± 5.1 | 36.2 ± 3.6 | 40.1 ± 4.6 | 32.7 ± 2.3 | 40.6 ± 11.2 | 40.4 ± 11.7 |
| | | CPT-Blk | 41.0 ± 1.5 | 43.9 ± 1.7 | **35.8 ± 2.2** | 39.3 ± 1.5 | 46.1 ± 1.8 | 33.2 ± 1.3 | 43.4 ± 6.5 | 43.6 ± 6.4 |
| | | CPT-Seg | **41.3 ± 2.6** | **48.2 ± 4.6** | 35.7 ± 2.5 | **42.6 ± 2.9** | **49.3 ± 4.7** | **35.4 ± 1.0** | **47.4 ± 3.5** | **47.4 ± 3.5** |
| | 16 | Fine-tuning | 39.8 ± 4.2 | 45.5 ± 5.0 | 34.9 ± 3.0 | 41.8 ± 3.0 | 47.3 ± 3.1 | 36.2 ± 2.3 | 47.5 ± 4.1 | 47.8 ± 4.7 |
| | | CPT-Blk | 44.8 ± 3.3 | 51.4 ± 4.1 | **38.2 ± 2.3** | 41.5 ± 1.3 | 48.2 ± 2.1 | 34.7 ± 0.9 | 47.8 ± 2.1 | 48.2 ± 2.8 |
| | | CPT-Seg | **45.3 ± 1.8** | **53.3 ± 3.0** | 37.5 ± 1.3 | **44.8 ± 0.9** | **52.5 ± 1.2** | **36.6 ± 1.2** | **51.0 ± 2.6** | **51.4 ± 2.8** |
| Fully Supervised | $|\mathcal{D}_{train}|$ | MAttNet | 76.7 | 81.1 | 70.0 | 65.3 | 71.6 | 52.0 | 66.6 | 67.3 |
| | | VL-T5 | – | – | – | – | – | – | 71.2 | 71.3 |
| | | ViLBERT | – | – | – | 72.3 | 78.5 | 62.6 | – | – |
| | | VLBERT | – | – | – | 71.6 | 77.7 | 61.0 | – | – |
| | | ERNIE-ViL | – | – | – | 74.0 | 80.3 | 64.7 | – | – |
| | | UNITER | 81.2 | 86.5 | 73.9 | **75.3** | **81.3** | **65.6** | 74.3 | 74.5 |
| | | Fine-tuning | 81.8 | **87.2** | 74.3 | 74.5 | 80.8 | 64.3 | 74.6 | **75.7** |
| | | CPT-Blk | 81.9 | 87.1 | 73.8 | 74.4 | 80.4 | 64.1 | **75.3** | 75.3 |
| | | CPT-Seg | **81.9** | 86.4 | **74.4** | 73.9 | 79.5 | 64.4 | 74.4 | 75.2 |

as the CPT backbone. We compare CPT with a series of strong baselines that utilize detected proposals, including vanilla fine-tuning of VinVL and other VL-PTMs (see Section D.1 for more baseline details). For fair comparisons, we adopt the base size for all VL-PTMs. We refer readers to Section A.1 for the results of large size VL-PTMs.

## 4.2 MAIN RESULTS

The main results are reported in Table 1, from which we observe that: (1) CPT outperforms the random baseline and the strong fine-tuning baseline by a large margin in zero-shot and few-shot settings. For example, using colored blocks as visual sub-prompts, CPT achieves 17.3% absolute accuracy improvement on average with one shot in RefCOCO evaluation. This indicates that CPT can effectively improve sample efficiency in tuning VL-PTMs. (2) Coloring objects with segmentation masks in visual sub-prompts (CPT-Seg) achieves even better results than blocks (CPT-Blk). The reason is that solid colors that fit the outlines of objects are more common in real-world images, making CPT-Seg more natural visual sub-prompts (despite requiring stronger annotation to train the segmentation tools). (3) Notably, CPT achieves significantly smaller standard deviation than fine-tuning. For example, CPT-Blk achieves 73.8% relative standard deviation reduction on average with one shot in RefCOCO evaluation. This shows that a coherent tuning approach from pre-training can lead to substantially more stable few-shot training, which is a crucial factor for evaluating few-shot learning models (Gao et al., 2021). (4) We note that CPT-Blk slightly underperforms fine-tuning with 16 shots in RefCOCO+ evaluation. The reason is that RefCOCO+ has more color-based expressions (e.g., *the person in red shirt and blue hat*), which can disturb our color-based CPT. However, this problem can be alleviated with more tuning instances in the fully supervised scenario, where models can learn to better distinguish colors in the query text and prompt template. (5) CPT models achieve comparable performance to strong fine-tuned VL-PTMs in the fully supervised settings. This shows that CPT is a competitive tuning approach for VL-PTMs even in the fully supervised scenario. We note that CPT-Blk slightly outperforms CPT-Seg in the fully supervised setting, and we refer readers to Section A.3 for a detailed analysis. In summary, compared to the vanilla fine-tuning approach, CPT achieves superior/comparable, and more stable performance in zero-shot, few-shot and fully supervised visual grounding.

## 4.3 INFLUENCE OF COLORS IN CPT'S VISUAL GROUNDING

In our analysis, we first investigate the influence of colors—the key ingredients—in the visual grounding performance of CPT. Specifically, we compare colors obtained from the frequency-based baseline (**Freq**) (See Section 3.4) and our cross-modal prompt search method CPS (**Ours**) in two dimensions, including an overall evaluation of top-N colors and a zoom-in study of individual colors. Unless otherwise specified, all the following experiments are conducted based on CPT-Blk on the validation set of RefCOCO in $0, 2, 8$ shot settings.

Table 2: Top-6 colors from the frequency-based baseline and our CPS. Visual appearances and color texts are reported. Best viewed in color.

| Model | Color #1 | Color #2 | Color #3 | Color #4 | Color #5 | Color #6 |
|-------|----------|----------|----------|----------|----------|----------|
| Freq | (255,0,0), red | (0,0,0), black | (0,0,255), blue | (0,255,0), green | (255,255,0), yellow | (165,42,42), brown |
| Ours | (240,0,30), red | (155,50,210), purple | (255,255,25), yellow | (0,10,255), blue | (255,170,230), pink | (0,255,0), green |

**Overall Evaluation of Top-N Colors.** We first show the top-6 colors recommended by each approach in Table 2. To evaluate the overall performance of the top colors from different models, we evaluate CPT equipped with each color from the top-6 colors respectively, and report the mean accuracy and standard deviation over different colors. From the experimental results in Figure 3a, we observe that the top colors produced by CPS achieve both higher mean accuracy and lower standard deviation than the baseline method in different shot-settings. The reason is that CPS jointly considers visual and textual semantics in searching cross-modal prompts, and therefore is able to effectively adjust and rank the colors for more accurate and stable visual grounding.

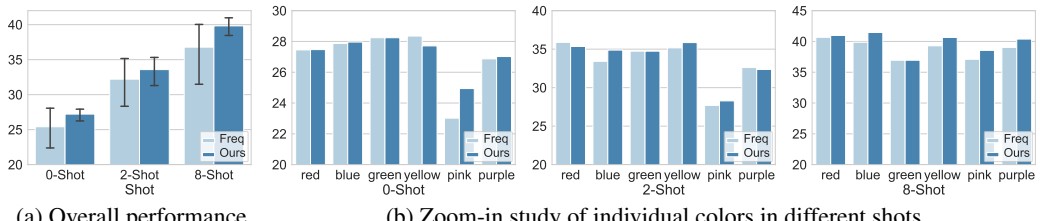

(a) Overall performance.                (b) Zoom-in study of individual colors in different shots.

Figure 3: Results of utilizing different colors for visual grounding, including (a) an overall evaluation of top-6 colors from different models, and (b) a zoom-in study of aligned individual colors.

**Zoom-In Study of Individual Colors.** To investigate the fine-grained influence of specific colors in CPT's visual grounding, we further perform a zoom-in study of individual colors. To align the colors for comparison, we merge the top-6 colors from the baseline and CPS, and remove the colors that are not included in the models' complete color sets (e.g., $black \notin \mathcal{C}$ in CPS). We report the accuracies in Figure 3b, from which we observe that: (1) The performance of different colors varies greatly in prompting VL-PTMs in the same shot-settings, and the optimal colors are different in different shot-settings. The results indicate the large influence of cross-modal prompt configurations, consistent with the findings from recent studies in textual prompt tuning (Jiang et al., 2020; Gao et al., 2021). (2) Colors produced by CPS achieve comparable or superior performance compared to the baseline in individual colors. The results show that given the color texts, CPS can properly adjust the color visual appearance (i.e., RGB) to improve the visual grounding performance. (3) We note that in some cases, colors produced by CPS slightly underperform the baseline. We hypothesize the reason is that, CPS uses a single textual template to compute the decoding scores for color adjustment, which can be biased. The problem can potentially be addressed by ensembling templates as in Qin & Eisner (2021), which we leave for future work.

## 4.4 CASE STUDY

To provide a more intuitive understanding of CPT, we conduct a case study on the validation set of RefCOCO in 8-shot setting. From the results in Figure 4, we have the following observations: (1) CPT enables VL-PTMs to distinguish target objects distracted by the same of type objects using

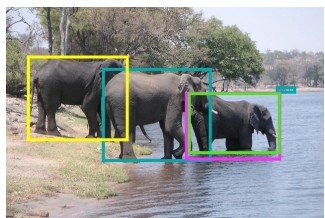
Query Text: right elephant in water

(a) Correctly predicted

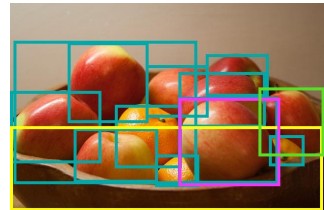
Query Text: apple on the bottom to the right of the orange in middle

(b) Disturbed by objects of the same type, but still reasonable

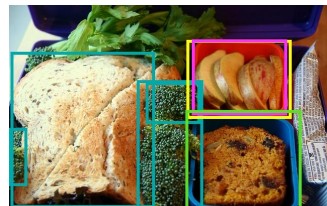
Query Text: food in red bowl

(c) Disturbed by colors in raw image regions and text

Figure 4: Case study. The bounding boxes given by image region proposals (olive), ground-truth annotation (pink), CPT (green), and fine-tuning baseline (yellow) are highlighted accordingly.

Table 3: Predicate classification results on Visual Genome. ZS: zero-shot, FS: fully supervised. We report the mean and standard deviation performance over 2 random splits.

| | Shot | Model | Val | | | | Test | | | |
|---|---|---|---|---|---|---|---|---|---|---|
| | | | R@50 | R@100 | mR@50 | mR@100 | R@50 | R@100 | mR@50 | mR@100 |
| ZS | 0 | Random | $1.6 \pm 0.2$ | $1.8 \pm 0.2$ | $1.1 \pm 0.2$ | $1.3 \pm 0.1$ | $1.5 \pm 0.0$ | $1.8 \pm 0.1$ | $1.2 \pm 0.1$ | $1.6 \pm 0.1$ |
| | | CPT-Blk | **33.6** | **34.7** | **14.8** | **15.5** | **29.3** | **30.5** | **13.0** | **14.5** |
| Few-Shot | 1 | Fine-tuning | $3.8 \pm 0.1$ | $4.2 \pm 0.1$ | $7.8 \pm 0.9$ | $8.7 \pm 1.0$ | $4.1 \pm 0.1$ | $4.7 \pm 0.0$ | $6.7 \pm 0.3$ | $7.6 \pm 0.4$ |
| | | CPT-Blk | $\mathbf{16.3 \pm 2.0}$ | $\mathbf{17.5 \pm 2.3}$ | $\mathbf{25.2 \pm 0.7}$ | $\mathbf{27.4 \pm 0.8}$ | $\mathbf{18.0 \pm 2.8}$ | $\mathbf{20.0 \pm 3.0}$ | $\mathbf{23.9 \pm 0.3}$ | $\mathbf{26.3 \pm 0.3}$ |
| | 4 | Fine-tuning | $7.1 \pm 1.9$ | $7.6 \pm 2.0$ | $10.3 \pm 0.8$ | $11.7 \pm 0.8$ | $7.3 \pm 1.5$ | $7.9 \pm 1.7$ | $11.8 \pm 1.0$ | $13.2 \pm 0.9$ |
| | | CPT-Blk | $\mathbf{14.4 \pm 0.4}$ | $\mathbf{15.4 \pm 0.4}$ | $\mathbf{30.4 \pm 1.5}$ | $\mathbf{32.8 \pm 1.6}$ | $\mathbf{17.7 \pm 0.6}$ | $\mathbf{19.3 \pm 0.6}$ | $\mathbf{28.5 \pm 1.5}$ | $\mathbf{32.1 \pm 1.0}$ |
| | 16 | Fine-tuning | $8.4 \pm 0.3$ | $8.9 \pm 0.3$ | $20.7 \pm 0.6$ | $21.7 \pm 0.6$ | $10.4 \pm 0.7$ | $11.2 \pm 0.8$ | $19.7 \pm 0.1$ | $21.7 \pm 0.1$ |
| | | CPT-Blk | $\mathbf{15.0 \pm 0.6}$ | $\mathbf{16.0 \pm 0.8}$ | $\mathbf{33.0 \pm 0.2}$ | $\mathbf{35.4 \pm 0.6}$ | $\mathbf{18.4 \pm 1.0}$ | $\mathbf{20.0 \pm 1.1}$ | $\mathbf{32.5 \pm 0.5}$ | $\mathbf{36.1 \pm 0.6}$ |
| | 32 | Fine-tuning | $9.7 \pm 1.1$ | $10.2 \pm 1.1$ | $21.9 \pm 0.6$ | $22.9 \pm 0.2$ | $11.7 \pm 0.2$ | $12.4 \pm 0.3$ | $22.0 \pm 0.1$ | $24.1 \pm 0.0$ |
| | | CPT-Blk | $\mathbf{17.2 \pm 0.4}$ | $\mathbf{18.2 \pm 0.4}$ | $\mathbf{34.6 \pm 0.2}$ | $\mathbf{37.9 \pm 0.1}$ | $\mathbf{20.8 \pm 0.1}$ | $\mathbf{22.3 \pm 0.1}$ | $\mathbf{34.0 \pm 0.1}$ | $\mathbf{37.7 \pm 0.3}$ |
| FS | $|\mathcal{D}_{\text{train}}|$ | Neural Motif | - | - | - | - | 65.2 | 67.0 | 14.8 | 16.1 |
| | | BGNN | - | - | - | - | 59.2 | 61.3 | 30.4 | 32.9 |
| | | PCPL | - | - | - | - | 50.8 | 52.6 | 35.2 | 37.8 |
| | | DT2-ACBS | - | - | - | - | 23.3 | 25.6 | 35.9 | 39.7 |

only a few training instances, while the fine-tuning method struggles to succeed (Figure 4a). (2) CPT can be distracted by hard candidates (e.g., objects of the same type as the target that requires complex reasoning to identify), but will typically produce reasonable predictions. For example, in Figure 4b, CPT predicts a nearby *apple* while the fine-tuning baseline predicts a *bowl*. The reason is that CPT maximally reuses the pre-trained parameters of VL-PTMs, which can help prevent outrageous predictions that typically happen in few-shot fine-tuning. (3) However, we find that CPT can be disturbed by colors in raw image regions and text. For example, it can be difficult for the model to identify a *red bowl* when the candidate regions are colored by red blocks (Figure 4c).

## 4.5 EXPERIMENTS ON PREDICATE CLASSIFICATION

To investigate the generalization capability of CPT, we evaluate CPT on predicate classification task.

**Experimental Settings.** (1) Datasets. We adopt the popular Visual Genome dataset (Krishna et al., 2017), which contains 50 visual relations. We refer readers to Section D.2 for the dataset details. (2) Evaluation protocols. Following previous works (Xu et al., 2017; Chen et al., 2019), we use recall@N (R@N) and mean recall@N (mR@N) as the evaluation metrics. During training, K labeled instances are provided for each relation. (3) Baselines. We adopt fine-tuning of VinVL as our most direct baseline model. Specifically, we feed the image regions and their categories into the model, and concatenate the visual hidden representations of the subject and object. Then the object pair representation is fed into a softmax classifier. All VL-PTMs are in base size. We also report the results of strong baselines that are tailored for the task, and are fully supervised with $315, 642$ labeled triplets, including Neural Motif (Zellers et al., 2018), BGNN (Li et al., 2021a), PCPL (Yan et al., 2020) and DT2-ACBS (Desai et al., 2021).

**Results.** From the results in Table 3, we observe that: (1) CPT significantly outperforms the random baseline and the strong fine-tuning baseline in zero-shot and few-shot settings. For example, using 32 shots, CPT achieves a strong mR@100 of 37.7%, outperforming fine-tuning by 13.6% absolute points, and closely approaching state-of-the-art fully supervised DT2-ACBS. This indicates that CPT can improve sample efficiency in tuning VL-PTMs. (2) We note that while the macro performance of CPT monotonically increases as the shot number grows, the micro performance drops first in 1- and 4-shot settings. This is due to the distribution gap between the balanced training set (i.e., K shot for each relation) and the long-tail test set. Since the relations in the pre-training corpora also follow a long-tail distribution, CPT can achieve a high starting point for micro performance.

## 5 RELATED WORK

**Pre-trained Vision-language Models.** Existing VL-PTMs can be roughly divided into three categories according to their pre-training objectives and architectures: (1) *Masked language modeling* based VL-PTMs are mainly pre-trained to recover the masked tokens (Lu et al., 2019; Su et al., 2019; Tan & Bansal, 2019; Li et al., 2020; Yu et al., 2021); (2) *Auto-regressive language modeling* based VL-PTMs model image and text tokens with Transformer decoders auto-regressively (Ramesh et al., 2021; Wang et al., 2021); (3) *Contrastive learning* based VL-PTMs are pre-trained to holistically match image-text pairs (Radford et al., 2021; Li et al., 2021b). Note that our Cross-modal Prompt Tuning (CPT) framework is orthogonal to VL-PTM design. In this work, without loss of generality, we focus on prompting masked language modeling based VL-PTMs due to their prevalence and superior performance, while applying CPT to other VL-PTMs is also applicable.

**Prompt Tuning for NLP.** Prompt tuning for pre-trained language models is a rapidly emerging field in NLP (Raffel et al., 2019; Brown et al., 2020; Liu et al., 2021). Originally designed for probing knowledge in pre-trained language models (Petroni et al., 2019), prompt tuning has now been extended to handle a variety of NLP tasks, including language understanding (Schick & Schütze, 2021a;b) and generation (Li & Liang, 2021). To facilitate prompt engineering, Shin et al. (2020) propose to automatically generate prompt templates via gradient-based search. Most related to our work are Tsimpoukelli et al. (2021); Zhou et al. (2021); Wang et al. (2021) that present textual prompt tuning for VL-PTMs, achieving promising results on some vision-language tasks. However, similar to existing works in NLP, they focus on prompt engineering in text, keeping images untouched, and therefore can only perform holistic implicit visual grounding. In comparison, to the best of our knowledge, CPT is the first cross-modal prompt tuning framework tailored for both image and text, and is capable of explicitly grounding natural language to fine-grained image regions.

**Visual Grounding.** There is a general consensus that visual grounding plays an essential role in solving vision-language tasks (Karpathy & Fei-Fei, 2015; Plummer et al., 2015; Goodfellow et al., 2016; Krishna et al., 2017; Lu et al., 2019). Mao et al. (2016) propose the referring expression comprehension task to explicitly evaluate the visual grounding capability. To address the task, most models learn to classify or rank image region candidates based on the expressions in a fully supervised fashion (Mao et al., 2016; Zhang et al., 2018; Lu et al., 2019; Chen et al., 2020), requiring large amounts of costly human-annotated data. To alleviate reliance on human annotation, some works have investigated zero-/few-shot grounding of new object types (Sadhu et al., 2019; Blukis et al., 2020), whereas amounts of training data are still needed for existing object types. In comparison, we prompt general VL-PTMs for zero- and few-shot visual grounding in a reformulated fill-in-the-blank paradigm independent of specific object types.

## 6 CONCLUSION AND FUTURE WORK

In this work, we present the first Cross-modal Prompt Tuning (CPT) framework for VL-PTMs. To facilitate prompt engineering, we present a principled approach to search for cross-modal prompt configurations. Comprehensive experimental results demonstrate the effectiveness of CPT on zero-shot, few-shot and fully supervised visual grounding. In future, we plan to address the color disturbance and improve the computation efficiency of CPT, and also investigate the effectiveness of CPT on other vision-language tasks. As the first attempt in cross-modal prompt tuning, we propose a color-based framework as one of the possible prompt tuning solutions. We leave exploring other plausible prompt tuning approaches of VL-PTMs for future work.

## 7 ETHICS STATEMENT

In this section, we discuss the main ethical considerations of CPT: (1) Intellectual property protection. The codes and data adopted from previous works are granted for research-purpose usage. (2) Privacy. The data adopted in this work (i.e., the pre-training data and tuning data) is created by human annotators for research purposes, and should not cause privacy issues. (3) Potential problems. VL-PTMs may be biased towards some objects and attributes. There are increasing efforts to address the problem in the community (Ross et al., 2021; Zhao et al., 2021).

## 8 REPRODUCIBILITY STATEMENT

To maximize the reproducibility, we provide a clear description of the methodology in Section 3, the pseudo-code of the model in Section B, implementation details in Section C, and detailed data characteristics and evaluation protocols in Section 4.1. All the data and codes will be available to facilitate future research.

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

# A SUPPLEMENTARY EXPERIMENTS

## A.1 RESULTS OF LARGE SIZE VL-PTMS

In this section, we report the experimental results of large size VL-PTMs, including vanilla fine-tuning of baseline VL-PTMs, CPT-Blk and CPT-Seg with large size backbone (i.e., $1,024$ dimensional hidden representations and $24$ layers). From the experimental results in Table 4, we observe that compared with vanilla fine-tuning, CPT achieves significantly better and more stable performance in zero-shot and few-shot settings, and comparable results in the fully supervised settings, which is consistent with the conclusions of main experiments in Section 4.2. In summary, the results show that CPT can generalize to VL-PTMs of different sizes.

Table 4: Results of large size VL-PTMs. Accuracies (%) of grounding referring expressions in zero-shot, few-shot and fully supervised settings. We report mean and standard deviation performance over 5 random splits. ZS: zero-shot. Blk: colored block, Seg: colored segmentation mask.

| | Shot | Model | RefCOCO | | | RefCOCO+ | | | RefCOCOg | |
|---|---|---|---|---|---|---|---|---|---|---|
| | | | val | testA | testB | val | testA | testB | val | test |
| ZS | 0 | Random | $15.9 \pm 0.2$ | $19.4 \pm 0.6$ | $13.4 \pm 0.4$ | $16.1 \pm 0.1$ | $13.3 \pm 0.6$ | $20.0 \pm 0.2$ | $18.8 \pm 0.4$ | $19.2 \pm 0.3$ |
| | | CPT-Blk | 25.7 | 25.4 | 27.0 | 25.9 | 25.8 | 25.7 | 32.9 | 32.6 |
| | | CPT-Seg | **29.5** | **30.6** | **28.7** | **28.8** | **30.3** | **27.4** | **34.6** | **34.8** |
| Few-Shot | 1 | Fine-tuning | $18.5 \pm 3.4$ | $13.7 \pm 4.8$ | $25.0 \pm 3.7$ | $23.0 \pm 6.5$ | $22.8 \pm 8.2$ | $23.6 \pm 4.5$ | $30.6 \pm 7.3$ | $31.5 \pm 7.4$ |
| | | CPT-Blk | $36.4 \pm 3.5$ | $39.1 \pm 4.3$ | $34.3 \pm 2.7$ | $34.4 \pm 3.8$ | $38.7 \pm 5.4$ | $31.2 \pm 2.5$ | $38.7 \pm 4.8$ | $38.7 \pm 4.6$ |
| | | CPT-Seg | $\mathbf{39.3 \pm 4.2}$ | $\mathbf{43.2 \pm 5.6}$ | $\mathbf{35.5 \pm 2.4}$ | $\mathbf{35.9 \pm 3.8}$ | $\mathbf{41.0 \pm 5.0}$ | $\mathbf{31.2 \pm 2.8}$ | $\mathbf{40.9 \pm 6.0}$ | $\mathbf{41.0 \pm 6.1}$ |
| | 2 | Fine-tuning | $23.4 \pm 3.5$ | $21.1 \pm 5.2$ | $26.7 \pm 4.5$ | $28.3 \pm 2.3$ | $30.1 \pm 5.3$ | $26.4 \pm 2.8$ | $33.1 \pm 8.3$ | $33.4 \pm 8.2$ |
| | | CPT-Blk | $38.3 \pm 2.9$ | $40.5 \pm 4.2$ | $35.3 \pm 1.2$ | $36.2 \pm 5.5$ | $41.1 \pm 7.6$ | $31.9 \pm 3.3$ | $40.6 \pm 5.9$ | $41.3 \pm 6.1$ |
| | | CPT-Seg | $\mathbf{41.4 \pm 1.5}$ | $\mathbf{45.8 \pm 3.6}$ | $\mathbf{36.6 \pm 2.0}$ | $\mathbf{38.7 \pm 3.8}$ | $\mathbf{44.7 \pm 5.2}$ | $\mathbf{33.5 \pm 2.6}$ | $\mathbf{43.2 \pm 5.9}$ | $\mathbf{43.4 \pm 5.8}$ |
| | 4 | Fine-tuning | $27.8 \pm 4.8$ | $26.0 \pm 7.8$ | $30.1 \pm 3.4$ | $33.4 \pm 3.5$ | $36.8 \pm 5.1$ | $28.3 \pm 2.1$ | $36.9 \pm 8.9$ | $37.2 \pm 8.7$ |
| | | CPT-Blk | $40.9 \pm 1.8$ | $45.0 \pm 2.0$ | $36.6 \pm 1.6$ | $37.2 \pm 3.6$ | $42.4 \pm 5.4$ | $33.6 \pm 2.3$ | $42.2 \pm 6.5$ | $42.7 \pm 6.9$ |
| | | CPT-Seg | $\mathbf{41.3 \pm 5.2}$ | $\mathbf{45.9 \pm 7.1}$ | $\mathbf{36.5 \pm 3.7}$ | $\mathbf{39.8 \pm 3.8}$ | $\mathbf{45.7 \pm 5.7}$ | $\mathbf{34.1 \pm 1.8}$ | $\mathbf{45.7 \pm 7.3}$ | $\mathbf{45.8 \pm 7.6}$ |
| | 8 | Fine-tuning | $33.3 \pm 4.2$ | $35.6 \pm 7.4$ | $31.2 \pm 2.7$ | $38.1 \pm 3.7$ | $43.5 \pm 3.9$ | $31.2 \pm 3.8$ | $41.9 \pm 8.0$ | $42.5 \pm 7.9$ |
| | | CPT-Blk | $42.7 \pm 4.1$ | $48.4 \pm 5.7$ | $37.3 \pm 2.4$ | $39.9 \pm 2.2$ | $45.8 \pm 3.0$ | $34.6 \pm 2.1$ | $44.8 \pm 4.1$ | $45.5 \pm 4.6$ |
| | | CPT-Seg | $\mathbf{45.2 \pm 3.6}$ | $\mathbf{51.4 \pm 4.9}$ | $\mathbf{38.7 \pm 2.4}$ | $\mathbf{42.4 \pm 3.8}$ | $\mathbf{49.0 \pm 4.9}$ | $\mathbf{35.7 \pm 1.8}$ | $\mathbf{48.1 \pm 5.4}$ | $\mathbf{48.6 \pm 5.8}$ |
| | 16 | Fine-tuning | $38.4 \pm 2.4$ | $42.8 \pm 4.2$ | $33.4 \pm 2.5$ | $40.7 \pm 3.2$ | $45.6 \pm 3.5$ | $34.7 \pm 2.8$ | $48.7 \pm 3.5$ | $49.4 \pm 3.5$ |
| | | CPT-Blk | $45.7 \pm 2.5$ | $53.0 \pm 3.2$ | $37.9 \pm 1.5$ | $41.8 \pm 2.0$ | $48.8 \pm 2.6$ | $35.7 \pm 1.4$ | $47.7 \pm 2.4$ | $48.6 \pm 2.8$ |
| | | CPT-Seg | $\mathbf{48.6 \pm 3.1}$ | $\mathbf{55.9 \pm 3.5}$ | $\mathbf{40.3 \pm 2.0}$ | $\mathbf{43.8 \pm 2.0}$ | $\mathbf{50.9 \pm 2.5}$ | $\mathbf{36.5 \pm 1.3}$ | $\mathbf{50.8 \pm 3.6}$ | $\mathbf{51.6 \pm 3.7}$ |
| Fully Supervised | $\|\mathcal{D}_{\text{train}}\|$ | MAttNet | 76.7 | 81.1 | 70.0 | 65.3 | 71.6 | 52.0 | 66.6 | 67.3 |
| | | ViLBERT | - | - | - | 72.3 | 78.5 | 62.6 | - | - |
| | | VLBERT | - | - | - | 72.6 | 78.6 | 62.3 | - | - |
| | | ERNIE-ViL | - | - | - | **76.0** | **82.1** | **66.9** | - | - |
| | | UNITER | 81.4 | 87.0 | 74.2 | 75.9 | 81.5 | 66.7 | **74.9** | **75.8** |
| | | Fine-tuning | 81.8 | **87.5** | 73.7 | 74.8 | 81.0 | 64.1 | 74.7 | **75.8** |
| | | CPT-Blk | 81.5 | 87.0 | **74.3** | 73.6 | 80.1 | 64.1 | 74.1 | 75.2 |
| | | CPT-Seg | **81.8** | 87.3 | 74.1 | 74.1 | 79.5 | 63.8 | 73.6 | 74.7 |

## A.2 EFFECT OF COLOR TRANSPARENCY

In practice, the color transparency is a crucial hyperparameter in CPT. Essentially, the choice of transparency is a trade-off between two factors: a small transparency can establish strong connections between color texts and visual appearances, but will undermine the visibility of the raw image region contents, and vice versa. To investigate the effect of color transparency, we evaluate CPT with different transparency of the default color (i.e., (240, 0, 30), *red*), with step size $0.1$ in grid search. From the results in Figure 5, we observe that: (1) The performance peaks at moderate transparencies in different shot-settings, which is consistent with our analysis of the trade-off. (2) Interestingly, the optimal transparency increases as the number of training shots grows. The reason is that the bottleneck of visual grounding in low shot

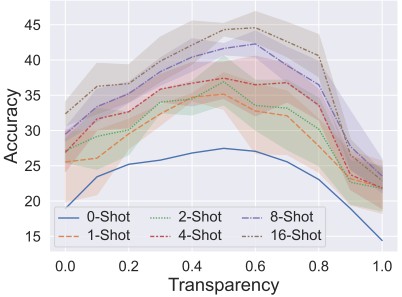

Figure 5: Experimental results with different color transparencies.

settings is to learn to utilize obvious colors in CPT to establish coarse-grained connections between

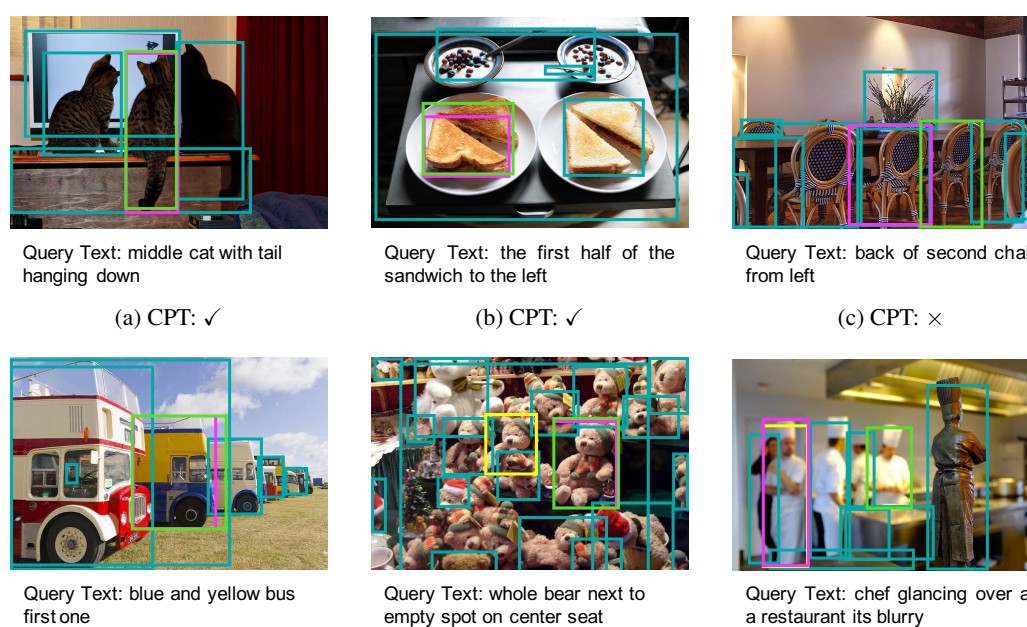

| | | |
|---|---|---|
| Query Text: middle cat with tail hanging down | Query Text: the first half of the sandwich to the left | Query Text: back of second chair from left |
| (a) CPT: ✓ | (b) CPT: ✓ | (c) CPT: × |
| Query Text: blue and yellow bus first one | Query Text: whole bear next to empty spot on center seat | Query Text: chef glancing over at a restaurant its blurry |
| (d) CPT: ✓, FT: ✓ | (e) CPT: ✓, FT: × | (f) CPT: ×, FT: ✓ |

Figure 7: Visualization of grounding results. First row: zero-shot setting. Second row: fully supervised setting. FT: fine-tuning. The bounding boxes given by image region proposals (olive), ground-truth annotation (pink), CPT (green), and fine-tuning baseline (yellow) are highlighted accordingly. Some images are cropped for better visual effects.

images and text. In comparison, in many shot settings, with a better mastery of colors, fine-grained reading and understanding of image regions become more important to handle hard instances that require complex reasoning (e.g., composition of attributes and relations).

### A.3 ANALYSIS OF VISUAL SUB-PROMPT SHAPE

In the main experimental results in Table 1, we note that although CPT-Seg significantly outperforms CPT-Blk in the zero-shot and few-shot settings, it slightly underperforms CPT-Blk in the fully supervised setting. To investigate the reason, we divide target objects in the validation set of RefCOCOg into disjoint bins according to the area of the bounding boxes, where each bin contains equal numbers of target objects (thus contributes equally to the overall result), and report the average performance of each bin in the fully supervised setting. From the results in Figure 6, we find that CPT-Seg outperforms CPT-Blk on large objects, but is inferior in grounding small objects. We hypothesize the reason is that CPT-Seg changes the object outlines with imperfect colored segmentation masks, hindering the understanding and reasoning of objects to some extent. The problem is exacerbated in small objects, since compared with large objects, the segmentation error of small objects is essentially enlarged when the object feature maps are pooled into input features of the same size for Transformers.

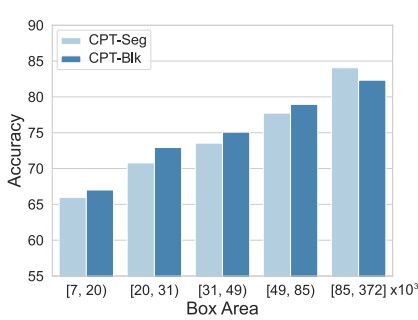

Figure 6: Performance of CPT-Blk and CPT-Seg with base size in different box areas in the fully supervised setting.

### A.4 VISUALIZATION

The few-shot grounding results are visualized in Figure 4. In this section, we further visualize the grounding results in zero-shot and fully supervised settings, as shown in Figure 7. We find that

CPT can make reasonable zero-shot predictions. Moreover, we observe that the color disturbance problem is largely alleviated in the fully supervised setting, i.e., CPT is less disturbed by colors in raw image and text, as shown in Figure 7d. The reason is that a capable VL-PTM can learn to largely distinguish the colors of varying objects and pre-defined maker blocks.

## B  Pseudo-Code of Cross-modal Prompt Search

Here we provide the pseudo-code of cross-modal prompt search. The algorithm aims to jointly consider visual and textual semantics in real-world cross-modal data to search for the color set $\mathcal{C}$ in CPT. The algorithm is simple in its design, and we leave exploring more advanced cross-modal prompt search methods for future work.

---

**Algorithm 1** Cross-modal Prompt Search

---

**Require:** $P(\cdot, \cdot)$: VL-PTM with image regions $\mathcal{R}$ and query
         text $q$ as input
**Require:** $\hat{\mathcal{C}}_w$: candidate color text set
**Require:** $\hat{\mathcal{C}}_v$: candidate color RGB set

1: **for** $\hat{c}_v^i$ in $\hat{\mathcal{C}}_v$ **do**
2:     $\mathcal{R}$ = {a pure color block of $\hat{c}_v^i$}
3:     $q$ = "`[CLS]` a photo of `[MASK]` color `[SEP]`"
4:     **for** $\hat{c}_w^j$ in $\hat{\mathcal{C}}_w$ **do**
5:        $s(\hat{c}_v^i, \hat{c}_w^j) = P(\texttt{[MASK]} = \hat{c}_w^j | \mathcal{R}, q)$
6:     **end for**
7: **end for**
8: Discard color candidates with low decoding scores
9: // Select sensitive color texts
10: $\mathcal{C}_w = \{c_w | c_w = \arg\max_{\hat{c}_w^j \in \hat{\mathcal{C}}_w} s(\hat{c}_v^i, \hat{c}_w^j), \hat{c}_v^i \in \hat{\mathcal{C}}_v\}$
11: // Select corresponding sensitive color RGB
12: $\mathcal{C} = \{(c_v, c_w) | c_v = \arg\max_{\hat{c}_v^i \in \hat{\mathcal{C}}_v} s(\hat{c}_v^i, c_w^j), c_w^j \in \mathcal{C}_w\}$

---

## C  Implementation Details

In this section, we provide the implementation details about model training and inference, object detection and segmentation, as well as cross-modal prompt search.

**Backbone.** We adopt the widely used VinVL (Zhang et al., 2021) as the backbone, which achieves strong performance on many vision-language tasks. We use the VinVL$_{\text{base}}$ model in the main experiments, with 768 dimensional hidden representations and 12 encoding layers.

**Object Detection and Segmentation.** During training and inference, we use the region proposals predicted by the Faster-RCNN (Ren et al., 2015) and object segmentation masks predicted by the Mask-RCNN (He et al., 2017), which are provided by MAttNet (Yu et al., 2018). Both Faster-RCNN and Mask-RCNN are based on ResNet101 (He et al., 2016) with a region proposal network and a fully connected classifier for object detection. For Mask-RCNN, an additional mask branch is added to conduct multi-task learning. The Faster-RCNN and Mask-RCNN provided by MAttNet (Yu et al., 2018) achieve 34.1 and 30.7 average precision on the COCO test set respectively.

**Visual Grounding.** During training, an image region is considered as the target if its intersection-over-union (IoU) with the ground-truth region is greater than $0.5$. During inference, we select the target region with the largest decoding score. All the hyperparameters and models are selected by grid search based on the performance on the few-shot/full validation set. The learning rate is $6e{-}5$, and decreases linearly towards $0$, which will be achieved at the end of the training ($500$ and $20,000$ steps for few-shot and fully supervised training respectively). The batch size is $32$ and identical to shot size in fully supervised and few-shot settings respectively. The size of image region batch is $1$.

**Predicate Classification.** The hyperparameters and models are selected by grid search on the validation set. The learning rate is 3e-5, and decreases linearly towards $0$, which will be achieved at the end of the training ($200$ steps). The batch size is identical to shot size.

Table 5: Relations in Visual Genome dataset (Krishna et al., 2017). Some relations are renamed (shown in parentheses) to better fit the query template.

| at | in | to | on | of | and | for | has (having) |
|----|----|----|----|----|-----|-----|--------------|
| says (saying) | over | from | with | near | wears (wearing) | under | above |
| using | along | behind | riding | across | eating | holding | wearing |
| between | against | playing | watching | carrying | covering | made of | part of |
| lying on | parked on | flying in | laying on | growing on | looking at | walking on | walking in |
| sitting on | covered in | mounted on | painted on | standing on | attached to | belonging to | hanging from |
| in front of | on back of | | | | | | |

**Cross-modal Prompt Search.** The color text candidate set $\hat{\mathcal{C}}_w$ is obtained from Wikipedia at `en.wikipedia.org/wiki/Lists_of_colors`. The color appearance candidate set $\hat{\mathcal{C}}_v$ is obtained by grid searching RGB candidates around the standard RGBs of color texts in $\hat{\mathcal{C}}_w$. In grid searching RGB candidates, the range is $\pm 30$ around standard RGBs with step size 5 in each channel. Colors candidates are discarded if the decoding score is less than $0.8$. The specific color choice from $\mathcal{C}$ is determined based on the performance on the few-shot/full validation set. The optimal color used in our experiments are $c = ((240, 0, 30), red)$ with transparency value $0.5$ in zero-shot and few-shot settings, and $c = ((255, 170, 230), pink)$ with transparency value $0.45$ in the fully supervised setting.

# D  EXPERIMENT DETAILS

## D.1  BASELINE DETAILS

We provide baseline details for visual grounding. (1) Vanilla fine-tuning for VinVL (Zhang et al., 2021). This model adopts the same backbone as CPT, and serves as the most direct baseline in few-shot and fully supervised experiments. Following Chen et al. (2020), the logits for all regions are fed into a softmax layer, and the score of the target region is optimized using cross-entropy objective. (2) Vanilla fine-tuning for other VL-PTMs. For fully supervised experiments, we also report previous results of fine-tuning other VL-PTMs, including ViLBERT (Lu et al., 2019), VLBERT (Su et al., 2019), UNITER (Chen et al., 2020), ERNIE-ViL (Yu et al., 2021) and VL-T5 (Cho et al., 2021). (3) Visual grounding model. MAttNet (Yu et al., 2018) is a strong model tailored for visual grounding, and is compared in fully supervised setting. (4) Random baseline. For zero-shot experiments, we compare with a random baseline that randomly guesses the target region. For fair comparisons, we use the object proposals detected by MAttNet (Yu et al., 2018) for all baselines and CPT-Blk.

## D.2  DATASET DETAILS

**Visual Grounding Datasets.** (1) RefCOCO (Yu et al., 2016) is collected through a two-player referential game (Kazemzadeh et al., 2014), and contains 142,210 referential expressions for 50,000 object instances in 19,994 images. The dataset is split into train, validation, testA and testB sets, with 120,624, 10,834, 5,657 and 5,095 expression-object pairs respectively. TestA set only contains people as target objects, while testB set contains all other types of objects as targets. (2) RefCOCO+ (Yu et al., 2016) is also collected in an interactive way, and contains 141,564 referential expressions for 49,856 object instances in 19,992 images. The difference from RefCOCO is that RefCOCO+ focuses on distinguishing objects using appearance-based expressions, and excludes location-based expressions. The dataset is split into train, validation, testA and testB sets, with 120,191, 10,758, 5,726 and 4,889 expression-object pairs respectively. (3) RefCOCOg (Mao et al., 2016) is collected in a non-interactive way, and contains 95,010 referential expressions for 49,822 object instances in 25,799 images. The referential expressions in RefCOCOg are typically longer and more complex. The train, validation and test sets contain 80,512, 4,896 and 9,602 expression-object pairs.

**Predicate Classification Datasets.** We provide the relations of Visual Genome dataset in Table 5. The dataset contains $65, 651$, $5, 000$ and $32, 422$ images in training, validation and test set respectively, where each image contains an average of $10.3$ objects and $4.8$ labeled relation instances. There are 150 distinct object categories and 50 relation categories in the dataset.

# E DISCUSSION AND OUTLOOK

In this section, we discuss the limitations of CPT and promising directions for future research.

**Limitations.** Despite its promising performance on visual grounding, we note that there are several limitations in CPT: (1) Color disturbance. CPT takes advantage of colors to bridge visual and textual semantics, by adding color-based sub-prompts in both images and text. As shown in Section 4.4, the color-based prompt can be disturbed by colors in raw images and text. (2) Computation efficiency. In our experiments, to maximally avoid color disturbance and account for the limited number of color candidates, we adopt small image region batch sizes. This means that a data instance needs to be fed into the model multiple times in order to obtain the result. We believe addressing these challenges are promising directions for improving CPT.

**Outlook.** In this work, we take visual grounding as a representative example to demonstrate the effectiveness of CPT. In fact, CPT can be easily adapted to other vision-language tasks. Here we discuss the promising directions, as illustrated in Figure 8. The visual and textual sub-prompts in CPT can well capture fine-grained object-level semantics for object-level tasks, such as: (1) Object classification. By coloring object proposals with visual sub-prompt, VL-PTMs can be prompted to produce object labels for object classification. (2) Scene graph classification. Moreover, by further decomposing textual sub-prompts, complex tasks involving different sub-tasks can be solved in a unified cross-modal prompt tuning framework. For example, VL-PTMs can be prompted to jointly produce object and predicate labels for challenging scene graph classification. In addition to data efficiency, a crucial advantage of using CPT is that the object/predicate labels can be produced from open-world vocabularies, instead of fixed label sets.

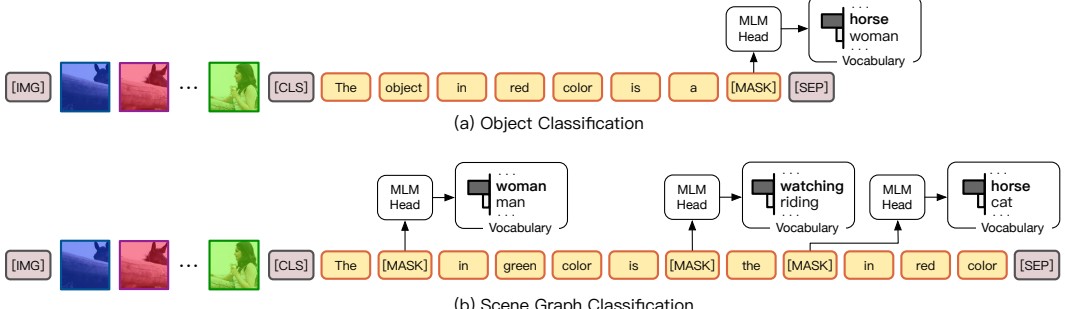

Figure 8: Outlook for adapting cross-modal prompt tuning (CPT) to other tasks.

