# OpenReview forum: "CPT: Colorful Prompt Tuning for Pre-trained Vision-Language Models"
_ICLR.cc/2022/Conference — ICLR 2022 Submitted_

### Official Review · Reviewer_vtFS · 2021-10-29

**Correctness:** 3
**Technical Novelty And Significance:** 3
**Empirical Novelty And Significance:** 3
**Recommendation:** 6
**Confidence:** 4

**Details Of Ethics Concerns:**

There are no ethical concerns.

**Main Review:**

Generally, CPT is an interesting, simple but effective prompt-based approach, which reformulates the problem to fit the downstream task. The experimental results and ablation studies are good and provide many insights. The paper is well written.

However, there are a few concerns of the paper:
1. I doubt that CPT is a generalizable approach that can fit other V+L tasks. Most V+L tasks do not need segmentation level information. Maybe the author can try some tasks like VCR. But for general task such as VQA, captioning, text-image retrieval, I do not think CPT can be applied. Therefore, CPT can not be claimed as a framework for VL-PTMs. It is only for visual grounding tasks.

2. The experiments only included a few simple baselines. We know the fully fine-tuning on large V+L models definitely can not generate reasonable results. It is possible to try some lightweight models like VL-T5 [1].
[1] Unifying Vision-and-Language Tasks via Text Generation

**Summary Of The Paper:**

This paper proposed CPT, colorful prompt tuning for visual grounding tasks using the pre-trained V+L model. By adding color-based co-referential markers in both image and text, CPT makes visual ground as a fill-in-the-blank problem and mitigates the gap between pre-training and fine-tuning. The experiments are conducted on three visual grounding tasks and demonstrate the effectiveness of CPT.

**Summary Of The Review:**

I think the paper is good and has the opportunity to be in ICLR. I would appreciate it if my concerns can be addressed in the rebuttal session.

-------------------------------------------------------------------------------------------------------------------
After reading the rebuttal, my concerns have been addressed partially. I will keep my original score.

---

> ### Author Response · Authors · 2021-11-20
> **Response to Reviewer vtFS**
>
> Thanks for the comments. We would answer the questions as follows:
>
> ### Generalization capability
> - We agree that it is important for CPT to generalize to other V+L tasks. Therefore, we further evaluate CPT on the task of predicate classification, which is quite a different task from visual grounding. In our revised version, we introduce the CPT approach for predicate classification in Section 3.5, and report the experimental results in Section 4.5. Experimental results show that CPT can also achieve competitive performance on predicate classification, which demonstrates the generalization capability of the model.
> - In general, we believe that CPT can also potentially facilitate other V+L tasks that do not require explicit modeling of the object locations, such as visual question answering. A promising approach will be first grounding objects in images, and then using the grounded image and text to perform downstream tasks, where both phases can be facilitated by CPT. For example, for visual question answering, given an image and a question “What is the man on the horse looking at?”, we can first ground “the man on the horse” in the image. Then the answer can be better predicted based on the explicitly grounded inputs in a fill-in-the-blank paradigm using CPT, such as “[CLS] What is the man on the horse in red color looking at? Answer: [MASK] [SEP]”. Some previous works have shown that explicit intermediate grounding can help improve not only the model performance, but also the robustness and interpretability for general V+L tasks [1,2].
>
> ### Lightweight baselines
> - Thanks for the suggestion. In fact, in our main experiments, all the VL-PTMs are in base size and have a comparable number of parameters with VL-T5. The description can be found in Section 4.1. Nevertheless, in our revised version, we further include VL-T5 in Table 1, which will be a good supplement for our experiments.
>
> ### Reference
> [1] Taking a HINT: Leveraging Explanations to Make Vision and Language Models More Grounded. CVPR 2019.
> [2] Multi-grained Attention with Object-level Grounding for Visual Question Answering. ACL 2019.

---

### Official Review · Reviewer_WQNe · 2021-11-02

**Correctness:** 3
**Technical Novelty And Significance:** 3
**Empirical Novelty And Significance:** 3
**Recommendation:** 6
**Confidence:** 4

**Main Review:**

[Strengths]
1.	The paper leverages the correspondence of colors between visual perception and textual input learned by the pre-trained model to enable visual grounding in downstream tasks, which is novel.
2.	Extensive ablation study and discussions are involved in the paper.
3.	The proposed Cross-Modal Prompt Search mines the most sensitive colors and considers the correspondence between visual and textual semantics to determine the color set used in the prompt template, which provides some insights of prompt search.

[Weaknesses]
1.	As discussed in Section 4.5, one limitation is that the added color masks would disturb the original colors in the input images and eventually mislead the model. It is believed that such cases are common in real-world data, such as “man in red shirt” or “white cat on the couch”.
2.	Image Region Batching would bring more cost during inference.
3.	The authors should visualize and compare the results of visual grounding derived by the proposed CPT and zero-shot/fine-tuning/fully supervised models.


**Summary Of The Paper:**

This paper proposes a novel paradigm named Cross-Modal Prompting Tuning (CPT) that reformulates visual grounding into a fill-in-the-blank problem. Specifically, CPT applies a unique colorful mask to each visual region in the input image and then utilizes a pre-defined template to wrap the input text, where the modal needs to identify the color of the corresponding region that contains the described object. Experiments are conducted on RefCOCO, RefCOCO+ as well as RefCOCOg and promising results are achieved.

**Summary Of The Review:**

Pls see the details in main review.

---

> ### Author Response · Authors · 2021-11-20
> **Response to Reviewer WQNe**
>
> We thank the reviewer for the comments. Here is our response to the concerns of the reviewer:
>
> ### Color disturbance and computation efficiency
> - We agree that fully addressing color disturbance and computation efficiency are important directions to improve CPT, as discussed in Section 4.5 and Section 5 in the initial version. Nevertheless, we observe that the color disturbance problem can be largely overcome by a capable VL-PTM even without taking any special measures. For example, for the 4,295 queries that have color texts on the validation set of RefCOCO+, where color disturbance happens, CPT can achieve a competitive 76.7% grounding accuracy, as compared with 77.1% grounding accuracy of vanilla fine-tuning baseline. The reason is that a capable VL-PTM can learn to largely distinguish the varying colors of objects and pre-defined colors of maker blocks.
>
> ### Computation efficiency of image region batching
> - Please refer to the response 1 above.
>
> ### More visualization results in different settings
> - Thanks for the suggestion. In the initial version, we visualize the results of CPT and the fine-tuning baseline in the few-shot setting in Section 4.5. In the revised version, we further provide the visualization results of CPT and the fine-tuning baseline in the zero-shot and fully supervised settings in Section A.4.

---

> > ### Comment · Reviewer_WQNe · 2021-11-29
> > **Additional comments after reading author response**
> >
> > First, thanks for your additional discussion and visualization results in the revised version, which indeed address most of my concerns.
> >
> > However, after checking the additional experiments of predicate classification, I notice that authors only report the results under the zero-shot and few-shot settings. It is better to include their results under fully-supervised setting.
> >
> > Moreover, I also agree with other Reviewers that authors should evaluate the proposed model in more common vision-language tasks (e.g., VQA). Otherwise, the title of "CPT: Colorful Prompt Tuning for Pre-trained Vision-Language Models" will mislead readers, since this work only support Visual Grounding (plus predicate classification in the revised version).

---

> > > ### Author Response · Authors · 2021-11-29
> > > **Response to Reviewer WQNe**
> > >
> > > Thanks for the comment. Here is our response:
> > > ### Fully supervised predicate classification results
> > > - We agree that it will be better to include the experimental results of fully supervised predicate classification. We will report the results in the final version.
> > >
> > > ### Evaluation on more general vision-language tasks
> > > - In the initial version, we show the effectiveness of CPT in locating image regions for ungrounded texts. In the revised version, we show that CPT can facilitate cross-modal tasks that aim to perform semantic recognition based on grounded inputs.
> > > - In general, we believe that CPT can also potentially facilitate other V+L tasks that do not require explicit modeling of the object locations, such as visual question answering. A promising approach will be first grounding objects in images, and then using the grounded image and text to perform downstream tasks, where both phases can be facilitated by CPT. For example, for visual question answering, given an image and a question “What is the man on the horse looking at?”, we can first ground “the man on the horse” in the image. Then the answer can be better predicted based on the explicitly grounded inputs in a fill-in-the-blank paradigm using CPT, such as “[CLS] What is the man on the horse in red color looking at? Answer: [MASK] [SEP]”. Some previous works have shown that explicit intermediate grounding can help improve not only the model performance, but also the robustness and interpretability for general V+L tasks [1,2]. We agree that further verifying CPT on visual question answering will make the paper stronger. The results will be included in the final version.
> > >
> > > ### Reference
> > > [1] Taking a HINT: Leveraging Explanations to Make Vision and Language Models More Grounded. CVPR 2019.
> > > [2] Multi-grained Attention with Object-level Grounding for Visual Question Answering. ACL 2019.

---

### Official Review · Reviewer_KcLt · 2021-11-07

**Correctness:** 3
**Technical Novelty And Significance:** 2
**Empirical Novelty And Significance:** 2
**Recommendation:** 5
**Confidence:** 4

**Main Review:**

Strengths:
- An interesting work to try to establish the connections with words and grounding image regions.
- It shows promising results on three refCOCO tasks.

Weakness:
There are some questions or missing parts -
1. What is the main pretrained model this work build on? VinVL? or ?

2. For these refCOCO datasets, what is the statistics for the language (text part) that covers the image regions? for example, on average, how many words in one sentence that covers the image regions? My understanding is for the high quality COCO captions, on average, there is 1~2 word objects that covers the image regions. This question may relate the motivation of this work, if the coverage is not too high, it might be quite trivial for this method.

3. Another question is - is it possible to verify this approach on a or more different task(s), rather than only refCOCO. For sure, refCOCO is a natural task for this method.

**Summary Of The Paper:**

This paper proposes a colorful prompt tuning for pre-trained vision-language models, with color masked regions and masked word tokens, on three reference tasks (refCOCOs), it shows promising results in zero-shot and few shot settings.

**Summary Of The Review:**

This is an interesting work to utilize the grounding between image regions and text words, as a prompt tuning way for pre-trained vision language models, and shows promising results on three refCOCO tasks in zero-shot and few-shot settings. One concern or question is to verify the generalization capability of this work in more broad settings or diverse tasks, or it is just task-specific for refCOCO.

Btw, I am not sure whether it is okay for section 8 & 9, which are on the page 10 (beyond 9 page limit).

---

> ### Author Response · Authors · 2021-11-20
> **Response to Reviewer KcLt**
>
> Thanks for the comments. We would answer the questions as follows:
>
> ### What is the main pre-trained model this work builds on? VinVL?
> - Yes, for the backbone of CPT, we adopt VinVL, which is a commonly used pre-trained model that achieves strong performance on many cross-modal tasks. The description can be found in Section B: Implementation Details in the initial version. For better readability, in the revised version, we have also added the description in the main text in Section 4.1.
> - In our main experiments, we report the experimental results of VinVL-base. To better demonstrate the effectiveness of CPT, we have also supplemented the results of VinVL-large in Section A.1 in the revised version, which shows consistent conclusions with base size model.
>
> ### Number of word objects that cover the image regions
> - Thanks for the comment. In fact, the main challenge of visual grounding is reasoning and locating the target object in a large number of image region candidates (potentially of the same type as the target object)[1,2]. The challenge remains even if there are only 1~2 word objects that cover the image regions. For example, given a picture of a crowd of people, and a query text “the tallest man”, there is at most 1 word object that covers the image region. The task will be challenging since a model needs to compare the heights of all men, and locate the tallest one. According to the statistics of RefCOCO test set, there are on average 36.0 image region candidates for each target object, among which 4.6 candidates are of the same type as the target, making the task rather challenging.
>
> ### Verification on more tasks
> - We agree that evaluating CPT on more tasks will make the paper stronger. Therefore, we further evaluate CPT on the task of predicate classification, which is quite a different task from visual grounding. In our revised version, we introduce the CPT approach for predicate classification in Section 3.5, and report the experimental results in Section 4.5. Experimental results show that CPT can also achieve competitive performance on predicate classification, which demonstrates the generalization capability of the model.
>
> ### Page limit
> - Thanks for the comment. According to the author guide, the ethics statement and reproducibility statement do not count towards the page limit.
>
> ### Reference
> [1] Modeling Context in Referring Expressions. ECCV 2016.
> [2] Grounding Referring Expressions in Images by Variational Context. CVPR 2020.

---

### Official Review · Reviewer_iZcJ · 2021-11-08

**Correctness:** 3
**Technical Novelty And Significance:** 3
**Empirical Novelty And Significance:** 3
**Recommendation:** 5
**Confidence:** 4

**Main Review:**

Strength:

1. The idea of connecting images and texts with the concept of color is quite neat. Color is probably one of the most prominent visual features but this seems to be the first time that it is used to connect the two modality.

2. CPT seems to tighten the gap between pretraining and fine-tuning. Previously, pretraining is a mask-language-modeling (MLM) task, while fine-tuning is a classification task that only uses the [CLS] head. Now, CPT also makes fine-tuning (prompt tuning) as a MLM task, so that it looks more similar to pretraining.

3. The studies on how the color can affect the performance is comprehensive.


Weakness and questions:

1. The method seems to rely on the quality of an image segmentation model to get the image regions. However, there is little detail about the model. How does the segmentation affect the CPT performance?

2. The query template shown in Figure 1 requires the main body of the query text is a noun. What if it is not that case? For example, what if the the query text is sentence “The horse is standing in the yard”?

3. The paper does not talk about how to ensure the MLM label in the cross-modal prompt tuning part (Figure 1(c)) is correct or how to deal with the ambiguity. For example, in Figure 1, how do you decide the answer is red or blue? For me, the woman can be watching either of the horses, so both red and blue can be correct.

4. The empirical evaluation is only on the visual grounding task, but there are quite a few other multimodal tasks that can be evaluated on to make the paper stronger, e.g., visual question answering. Actually, the authors have pointed out that the method can potentially be applied to object detection, predicate classification and scene graph classification in Figure 5. But why don’t you just work on those tasks to show CPT’s superiority? Furthermore, even for the visual grounding task, CPT is only evaluated on RefCOCO and its variants, making the claims less convincing. Minor question: why is there no CPT-Seg result in the last row of Table 1?


**Summary Of The Paper:**

This paper proposes a colorful prompt tuning (CPT) method for tuning pretrained vision-language models. CPT reformulates visual grounding into a fill-in-the-blank problem with color-based coreferential markers in image and text. The grounding to the target image region is achieved by recovering the corresponding color text from the masked token in the query template. Empirical studies show that CPT can outperform existing methods in promoting VL-PTMs for visual grounding in zero-shot, few-shot and fully supervised settings.

**Summary Of The Review:**

While the idea of this paper is neat, there are a few details missing. The empirical studies lack comprehensiveness to support that the proposed method can generalize. I would be more than happy to change my score if the concerns are addressed well.

---

> ### Author Response · Authors · 2021-11-20
> **Response to Reviewer iZcJ**
>
> We thank the reviewer for the detailed comments. Here is our response to the concerns of the reviewer:
>
> ### Details about object detection and segmentation models
> - **Model details.** In our experiments, all the baselines and CPT-Blk use the object bounding boxes predicted by the Faster-RCNN [1], and the object segmentation masks predicted by the Mask-RCNN [2]. The description can be found in Section B: Implementation Details in the initial version, and we have provided more details about the detection and segmentation models in the updated version in Section C.
> - **How does the segmentation affect the CPT performance.** In Section 4.2, we observe that coloring objects with segmentation masks will produce better results than blocks due to their natural fit to object outlines. In the revised version, we further analyze the influence of object detection/segmentation in Section A.3. We find that imperfect segmentations will hinder the grounding of small objects. Therefore, we expect improved object detection/segmentation models will lead to better performance of CPT.
>
> ### Main body of the query text is a noun
> - In the literature, the query text for visual grounding is commonly defined as a noun or noun phrase [3,4]. The rationale of this natural definition is that the task of grounding a whole sentence can be ambiguous. For example, given a query text “The horse is standing in the yard”, it is ambiguous whether the object to be grounded is the “horse” or the “yard”.
> - Nevertheless, CPT can be easily adapted to ground objects in a sentence, if the object word to be grounded is given. For example, we can simply change the template to “[CLS] *q*, where *o* is in [MASK] color [SEP]”, where *q* is the query text, and *o* is the noun to be grounded. For the above example, the input text will be: “[CLS] The horse is standing in the yard, where the horse is in [MASK] color [SPE]”. We agree it will be interesting directions for our future explorations.
>
> ### How to ensure that the label is correct
> - Generally, in our experiments, we use the human annotation from high-quality datasets to supervise and evaluate our model. Specifically, the target color text of the MLM head is the color of the ground-truth image region according to the RefCOCO annotation, which ensures that the label is correct.
> - For the specific example in Figure 1, it is indeed a hard grounding example. However, by observing the orientation of the woman’s eyes, face and right arm, it can be concluded that the horse in red color is closer to the correct answer.
>
> ### Evaluation on diverse tasks and datasets
> - **Evaluation on diverse tasks.** We agree that evaluating CPT on more tasks will make the paper stronger. Therefore, we further evaluate CPT on the task of predicate classification, which is quite a different task from visual grounding. In our revised version, we introduce the CPT approach for predicate classification in Section 3.5, and report the experimental results in Section 4.5. Experimental results show that CPT can also achieve competitive performance on predicate classification, which demonstrates the generalization capability of the model.
> - **Evaluation on diverse datasets.** Following previous works [5,6], we evaluate CPT on RefCOCO, RefCOCO+ and RefCOCOg, which are the most commonly used benchmarks for visual grounding, and exhibit diversity in multiple aspects. First, RefCOCO and RefCOCO+ are collected through a two-player game, while RefCOCOg is collected in a non-interactive way. Second, RefCOCO and RefCOCOg include both location-based and appearance-based expressions, while RefCOCO+ focuses on appearance-based expressions. Third, expressions in RefCOCO and RefCOCO+ are typically short, while RefCOCOg has much longer and more complex expressions.
> - **CPT-Seg in the last row of Table 1.**
> Thanks for the comment. The results are supplemented in Table 1 in the revised version.
>
> ### Reference
> [1] Faster R-CNN: Towards real-time object detection with region proposal networks. NeurIPS 2016.
> [2] Mask R-CNN. ICCV 2017.
> [3] Modeling Context in Referring Expressions. ECCV 2016.
> [4] Generation and Comprehension of Unambiguous Object Descriptions. CVPR 2016.
> [5] UNITER: Universal image-text representation learning. ECCV 2020.
> [6] Grounding Referring Expressions in Images by Variational Context. CVPR 2020.

---

> > ### Comment · Reviewer_iZcJ · 2021-11-29
> > **Predicate classification results**
> >
> > Thanks for adding the predicate classification results. Why are the zero-shot results of R@50 and R@100 on both Val and Test much better than few-shot, which is quite counter intuitive?

---

> > > ### Author Response · Authors · 2021-11-29
> > > **Response to Reviewer iZcJ**
> > >
> > > Thanks for the comment. As discussed in the experimental result analysis in Section 4.5, we note that while the macro performance of CPT monotonically increases as the shot number grows, the micro performance drops first in 1- and 4-shot settings. This is due to the distribution gap between the balanced training set (i.e., K shot for each relation) and the long-tail validation and test sets. Since the relations in the pre-training corpora also follow a long-tail distribution, CPT can achieve a high starting point for micro performance in the zero-shot setting.

---

> > > > ### Comment · Reviewer_iZcJ · 2021-11-29
> > > > **Thanks for the pointer**
> > > >
> > > > Thanks for the pointer and explanation. That being said, the predicate classification task is not a very widely used one among the popular V+L benchmarks. It seems this paper is reluctant to run the proposed model on the mainstream tasks such as VQA and image captioning (though it promises to do so), but evaluates only on some less common ones, making it hard to compare with state-of-the-art methods.
> > > >
> > > > Overall I appreciate the neat idea of this paper, but I am not sure if it is really better than the state-of-the-art vision-language models from the experiments. Therefore, I will keep my score unchanged.

---

> > > > > ### Author Response · Authors · 2021-11-30
> > > > > **Response to Reviewer iZcJ**
> > > > >
> > > > > Thanks for the comment. For the experiments, we compare our model with vanilla fine-tuning of the state-of-the-art VinVL model [1]. Experimental results show the superiority of our model over the state-of-the-art baseline. For example, our model outperforms the baseline by 17.3% accuracy in visual grounding, and 15.3% R@100 in predicate classification with one shot. We agree it will be helpful to further evaluate the generalization capability of the proposed model on visual question answering, which will be included in the final version.
> > > > >
> > > > > [1] Revisiting Visual Representations in Vision-Language Models. CVPR 2021.

---

### Author Response · Authors · 2021-11-20
**General Response**

We would like to thank all reviewers and ACs for their valuable time and efforts. We have taken the suggestions seriously and tried to address each concern. In addition to the detailed response to each reviewer, we also provide a general response here to help track our modifications that may be of general interest.

### Verifying the generalization capability on other tasks
- In our initial version, we show the effectiveness of CPT in visual grounding. The major concern of the reviewers is the generalization capability of CPT to other tasks, where three reviewers suggest further verifying CPT on other tasks to make the paper stronger. Therefore, we further evaluate CPT on the task of predicate classification, which is quite a different task from visual grounding. In our revised version, we introduce the CPT approach for predicate classification in Section 3.5, and report the experimental results in Section 4.5. Experimental results show that CPT can also achieve competitive performance on predicate classification, which demonstrates the generalization capability of the model.

### Experimental results of CPT with large size VL-PTM
- In the initial version, we adopt VinVL-base as the backbone. To further verify the generalization capability of CPT to large Vision-language Pre-trained models (VL-PTMs), we evaluate CPT with the VinVL-large backbone. We report the experimental results in Section A.1 in the revised version, which show consistent conclusions with the base size backbone.

### More visualization results
- In our initial version, we visualize the results of CPT and the fine-tuning baseline in the few-shot setting in Section 4.5. In the revised version, we further provide the visualization results of CPT and the fine-tuning baseline in the zero-shot and fully supervised settings in Section A.4.

---

### Author Response · Authors · 2021-11-27
**General Response**

Dear Reviewers,

Thanks for your reviews and suggestions. We have posted our responses to the concerns, as well as the revised paper. Could you please let us know if you have any further questions since we still could have interactions? We will respond as soon as possible.

Thanks, Authors

---

### Decision · Program_Chairs · 2022-01-20

**Decision:**

Reject

**Comment:**

In my opinion, this is a cool idea, but could use a few more test settings to evaluate the general applicability of their method. It would be interesting to see if the method generalizes to a non-reference based task.

Strengths:
Novel method that explores the interaction of color masks for learning to prompt about regions in images by identifying the color region they correspond to
Paper contains extensive ablation studies & discussions

Weaknesses:
Experimental results are run on uncommon benchmarks, making it difficult to compare to SOTA V+L methods
Consequently, it’s not clear that this method would generalize beyond visual grounding to tasks such as VQA or captioning